# Wild raccoons (*Procyon lotor*) as a potential reservoir of cytolethal distending toxin-producing *Providencia* strains in Japan

Okechukwu John Obi,[1] Atsushi Hinenoya,[1,2,3,4] Sharda Prasad Awasthi,[1,2,3,4] Noritoshi Hatanaka,[1,2,3,4] Shah M. Faruque,[5] Shinji Yamasaki[1,2,3,4]

**ABSTRACT**    In view of increasing reports of infections due to virulent *Providencia* species including cytolethal distending toxin (*cdt*) gene-positive strains, it is important to identify the reservoirs and transmission routes of such pathogenic strains. Raccoons considered to be a source of zoonotic pathogens were monitored for the presence of *Providencia* species in Japan and analyzed for *cdt* genes. Of 384 wild raccoon rectal swabs analyzed, 60% were positive for *Providencia* species, of which 20% carried *cdt*-genes. Among seven *Providencia* species isolated (*P. alcalifaciens, P. rustigianii, P. rettgeri, P. stuartii, P. heimbachae, P. vermicola,* and *P. huaxiensis*), *cdt* genes were distributed in *P. alcalifcaiens* (63%), *P. rustigianii* (16%), and novel in *P. rettgeri* (21%). Complete *cdt* gene clusters were identified in *P. alcalifaciens* and *P. rustigianii* strains, whereas *P. rettgeri* had intact *cdtB* but truncated *cdtA* and *cdtC* genes. Phylogenetic analyses showed divergent pulsotypes among the *cdt* gene-positive *Providencia* strains. Cytotoxicity assay revealed that *P. alcalifaciens* and *P. rustigianii* produced CDT more toxic to eukaryotic cells compared to human clinical strains, which were neutralized by anti-PaCdtB serum. As expected, the *P. rettgeri* strains with truncated *cdt* genes had no biological activity. Molecular analysis revealed that all the *cdt* genes were located on plasmids as determined by S1-nuclease pulsed-field gel electrophoresis (S1-PFGE) and Southern hybridization assay. Intriguingly, the *cdtB* gene in *P. rustigianii* strains was detected on dual plasmids. Notably, all the *cdt* gene-positive *Providencia* strains were found to carry plasmid-mediated T3SS-related genes. These results suggest that wild raccoons are possible reservoir of virulent *Providencia* strains in Japan.

**IMPORTANCE**    *Providencia* species considered normal flora are occasionally associated with gastroenteritis in healthy humans. Cytolethal distending toxin (CDT), a bacterial virulence factor found in various Gram-negative bacteria and associated with gastroenteritis and extra-intestinal infection has also been reported in at least two *Providencia* species (*P. alcalifaciens* and *P. rustigianii*). Determination of the transmission routes of such virulent *Providencia* is crucial for the implementation of evidence-based control programs. In this study, we identified raccoons as the probable reservoir of the *cdt* gene-positive *Providencia* strains in Japan. Interestingly, CDTs produced by raccoon-derived *Providencia* strains exerted more toxic effects on the eukaryotic cells compared to the clinical *Providencia* strains. In addition, the identification of a novel *cdt* gene cluster in another species *P. rettgeri* isolated from raccoons suggests that *Providencia* may be categorized as an emerging zoonotic pathogen.

**KEYWORDS**    cytolethal distending toxin (CDT), *Providencia*, *Providencia alcalifaciens*, *Providencia rustigianii*, *Providencia rettgeri*, raccoons, virulence

**Peer Reviewer** Thandavarayan Ramamurthy, National Institute of Cholera and Enteric Diseases, Kolkata, India

Address correspondence to Shinji Yamasaki, yshinji@omu.ac.jp, or Atsushi Hinenoya, hinenoya@omu.ac.jp.

The authors declare no conflict of interest.

See the funding table on p. 15.

The genus *Providencia* belongs to the Morganallaeceae family of Gram-negative bacteria formerly classified as Enterobacteriaceae (1). Currently, 11 species are recognized as members of the genus including *P. alcalifaciens, P. burhodogranariea, P. heimbachae, P. huaxiensis, P. mangonoxydans, P. rettgeri, P. rustigianii, P. sneebia, P. stuartii, P. thailandensis,* and *P. vermicola* (http://www.bacterio.net/providencia.html). *Providencia* species are attracting public attention due to increasing reports associating them with intestinal/extra-intestinal diseases (2–4).

*P. alcalifaciens* was associated with several outbreaks of foodborne illness in Japan, the Czech Republic, and Kenya (5–7). Species including *P. alcalifaciens, P. rettgeri, P. rustigianii, P. heimbachae,* and *P. vermicola* have been associated with diarrheal cases in humans and animals (2, 3, 8–12). *P. huaxiensis* was recovered from a human rectal swab in China (13) with a potential role in diarrhea. *P. stuartii* and *P. rettgeri* are frequently attributed to urinary tract infections in immuno-compromised and catheterized patients (14, 15). Furthermore, strains of the *Providencia* genus have been frequently isolated from wound infections, human blood, respiratory tract, poultry, and reptile feces (16). Importantly, the clinical relevance of the *Providencia* species also lies in their extensive antimicrobial resistance.

The precise virulence mechanisms of *Providencia* are not fully understood, however, motility, ability to adhere and invade eukaryotic cells (8, 17), and production of cytolethal distending toxin (CDT) (18) are considered the probable virulence factors. We have recently found that a type 3 secretion system (T3SS) co-localized with *cdt* genes on a conjugative plasmid distinct from the widely distributed chromosomal T3SSs is linked to invasion of human epithelial cells and diarrheagenicity of *P. rustigianii* (19). Virulence mechanism of CDT produced by *Providencia* spp. remains to be characterized, but the association of CDT-producing bacteria from several genera with inflammation and diarrhea in animal models and probably humans has been suggested by previous studies (20–23). CDT is a tripartite-holotoxin consisting of three subunits CdtA, CdtB, and CdtC acting in tandem to produce a functional toxin. CdtA and CdtC are considered the receptor-binding subunits, whereas CdtB is the effector domain with DNase I-like activity translocated into the cell with resultant cell cycle arrest, irreversible cell distension, and subsequent cell death (24). CDT has been proposed as a tri-perditious toxin that alters host defenses by disrupting epithelial cells, suppressing adaptive immunity, and promoting pro-inflammatory responses (25). Species-specific CDTs have been reported in several pathogenic Gram-negative bacteria including *Escherichia coli*, *Escherichia albertii, Campylobacter* spp., *Shigella* spp., *Salmonella enterica, Haemophilus ducreyi, Helicobacter* spp., and *Aggregatibacter actinomycetemcomitans* (18, 24, 26, 27). In addition, we have previously reported the presence of novel *cdt* gene clusters in clinical strains of *P. alcalifaciens* (18, 22) and *P. rustigianii* (3, 28), both from pediatric diarrhea cases in Japan.

Despite our findings of CDTs in the two *Providencia* species and reported findings by others in *P. alcalifaciens* elsewhere (29, 30), the likely reservoir and transmission route of such strains to humans has not been adequately explored. Previous (2, 31) and ongoing surveillance of *Providencia* spp. in human diarrhea samples in Japan, Thailand, and elsewhere ( O.J. Obi., et al unpublished data) yielded no significant indication of the probable source. Furthermore, we did not find any *cdt* gene-positive *Providencia* spp. in the surveillance of retail meats (beef, pork, chicken) in Japan, China, and Thailand (31, 32, Obi OJ et al., unpublished data). To expand our search, we conducted an exploratory survey in wild raccoons (*Procyon lotor*) captured as invasive animals by Department of Environment, Agriculture, Forestry and Fisheries Animal Proteciton and Livestock Division in Osaka Prefecture, and they were found to harbor *cdt* gene-positive *Providencia* strains. This finding prompted us to conduct a more extensive survey of the wild raccoons for the virulent (*cdt* gene-positive) *Providencia* species.

Raccoon (a synanthrope) is a known reservoir of some bacterial and viral pathogens including *Salmonella* serovars, *Campylobacter* spp., *E. albertii*, rabies virus, etc. (33–35). Raccoons are omnivorous and are known to adapt to diverse environments such as

farms, forests, and urban areas where they can forage agricultural crops and invade native animal species. In addition, they are thought to have a high potential for zoonoses transmission in this area. Hence there is a government policy to reduce their population through controlled extermination. To the best of our knowledge, this is the first report on the molecular epidemiology of *Providencia* spp. in a wild animal with the aim of ascertaining the probable reservoir of *Providencia cdt* gene-positive strains including their zoonotic relevance.

## RESULTS

### Prevalence of *Providencia* species and *Providencia cdt* genes in the raccoons

To examine the distribution of *Providencia* spp. in the wild raccoons, rectal swabs from 384 raccoons captured as invasive animals in Osaka Prefecture, Japan between November 2020 and April 2023 were evaluated using a duplex PCR developed in this study for the specific detection of *Providencia* genus-16S rDNA (16S rRNA) and their *cdt* (*Pcdt*) genes, respectively. The genus-specific 16S rRNA of *Providencia* was detected in 232 (60%) of the samples, and among them, 46 (20%) were also positive for the *Pcdt* genes (Table 1). *Providencia* species were isolated from the samples using a *Providencia*-specific isolation medium PMXMP (9), and the isolates were determined to be various species (Table 2) by conventional biochemical tests (Table S1), and sequencing of 16S rRNA and *rpoB* genes (Table S2). Mixed *Providencia* species were isolated from 36 raccoons; however, multispecies *cdt* gene-positive *Providencia* was not co-isolated from any of the raccoons. The *cdt* gene-positive *Providencia* species isolated from 30 samples were confirmed as *P. alcalifaciens* ($n = 24$), and *P. rustigianii* ($n = 8$).

We further assessed the possible presence of other CDT variants which cannot be detected by the duplex PCR assay, in the *cdt* gene-negative *Providencia* strains using a PCR-restriction fragment length polymorphism (RFLP) assay capable of amplifying and distinguishing *cdtB* genes of *E. coli*, *E. albertii*, and *Providencia* species (22). Interestingly, *P. rettgeri* strains isolated from eight samples yielded amplicons corresponding to the *cdtB* gene in size, but with different restriction patterns from those of the known *Providencia cdtB* (Table 2; Fig. S1 and S2). Together, *cdt* gene-positive *Providencia* was most frequently detected in *P. alcalifaciens* (63%), followed by *P. rettgeri* (21%), and *P. rustigianii* (16%) (Table 2).

### Genomic diversity of *cdt* gene-positive *Providencia* strains

To determine the phylogenetic relationship among the *cdt* gene-positive *Providencia* strains, one to two representative strains per *cdt* gene-positive sample were selected at random, and pulsed-field gel electrophoresis (PFGE) was carried out using *Not*I or *Sma*I digested genomic DNAs. *Not*I was chosen as the primary restriction enzyme due to its acceptable restriction patterns of 10–30 bands (36). Alternative restriction enzyme *Sma*I was used for the *P. alcalifaciens* strains due to the inability of *Not*I to digest most of the strains' genomes. Divergent pulsotypes consisting of 20, 4, and 3 distinct clones were observed in the *cdt* gene-positive *P. alcalifaciens*, *P. rettgeri*, and *P. rustigianii* strains, respectively (Fig. 1A through C). Furthermore, multiclonal *cdt* gene-positive *Providencia* strains were obtained from some of the raccoons including the raccoons tagged RAC2362 and RAC2369 (Fig. 1A through C).

**TABLE 1** Prevalence of *Providencia* species and *Providencia cdt* genes in the feces obtained from raccoons

| Target genes | No. of PCR-positive | No. of samples from which *Providencia* spp. were isolated |
|---|---|---|
| 16S rRNA | 232/384 (60%) | 230/384 (99%) |
| 16S rRNA and *cdt* | 46/232 (20%) | 38/46 (83%)[a] |

[a]Only *cdt* gene–negative *Providencia* were isolated in eight samples indicating probable mixed infections of *cdt* gene-positive and -negative *Providencia*.

**TABLE 2** Distribution of *Providencia* species in the feces of raccoons and of *Providencia cdt* genes in the isolates

| Species | Isolation frequency (%)[a] | No. of samples from which *cdt* genes-positive strains were obtained (%) | % *cdt* genes distribution among *cdt* genes-positive strains |
|---|---|---|---|
| *P. alcalifaciens* | 148/230 (64) | 24 (16) | 63 |
| *P. rustigianii* | 8/230 (3.5) | 6 (75) | 16 |
| *P. rettgerii* | 71/230 (31) | 8 (11)[b] | 21 |
| *P. stuartii* | 22/230 (10) | 0 (0) | 0 |
| *P. vermicola* | 5/230 (2.2) | 0 (0) | 0 |
| *P. heimbachae* | 8/230 (3.5) | 0 (0) | 0 |
| *P. huaxiensis* | 4/230 (1.7) | 0 (0) | 0 |

[a]Mixed *Providencia* species were isolated from 36 samples.
[b]*cdt* genes were negative by the duplex PCR, but positive by the alternative PCR-RFLP assay.

## Expression of CDT in the *Providencia cdt* gene-positive strains

To confirm whether the *cdt* gene-positive raccoon strains produce CDT, anti-PaCdtB serum (18) was used in assessing the presence of CdtB in their cell lysates by Western blotting (Fig. 2). A single band, corresponding to the expected size of CdtB (28 kDa) (28) was obtained from all the *cdt* gene-positive *P. alcalifaciens* and *P. rustigianii* strains, suggesting that CDT is produced by all the *cdt* gene-positive strains except for *P. rettgeri* (data not shown). Notably, highly intense CdtB bands were observed in the *P. rustigianii* strains compared to the other tested *Providencia* strains (Fig. 2). No such reactive bands were seen in the negative control strains (Lanes 2 and 3 in Fig. 2).

## Biological activities of the *Providencia* CDTs

Having seen the expression of CdtB in the raccoon-derived *cdt* gene-positive *Providencia* strains, we sought to examine the cytotoxic activities of the *Providencia* CDTs in eukaryotic cell lines. Sonicated filter-sterilized cell lysates of the *cdt* gene-positive *P. alcalifaciens* and *P. rustigianii* strains caused extended cell distension when incubated at 37°C for 72 h in CHO, Caco-2, and HeLa cells (Fig. 3; Table 3). The titer of CDT activities in each species varied considerably in each cell line. Culture supernatants of the strains in brain heart infusion revealed that cytotoxic titer in CHO cells was lower than that by sonicated filter-sterilized cell lysates (data not shown), indicating that CDT is produced in cell-associated fraction rather than supernatant. However, no cell distensions in the eukaryotic cell lines were seen with the *cdt* gene-positive *P. rettgeri* strains (Fig. 3; Table 3). Unexpectedly, we observed that the raccoon-derived strains had higher toxic titer to the eukaryotic cell lines compared to the clinical strains (Table 3). Intriguingly, cell lysates from the *P. rustigianii* strains caused higher cell toxicity in the eukaryotic cell lines compared to the *P. alcalifaciens* strains (Table 3).

To further examine whether the observed cytotoxicity in the eukaryotic cell lines is specific to *Providencia* CDT, CHO cells were incubated with the filter-sterilized cell lysates or culture supernatants in the presence of anti-PaCdtB rabbit serum. Indeed, the cell distension activities were neutralized by the addition of the antiserum suggesting that the observed distensions were due to CDT activity (Fig. 3).

## Nucleotide sequences of the *cdt* genes in the raccoon-derived *Providencia* strains

Based on the observed differences in the CDT biological activities and CdtB expression, and to gain further insights into the novel *cdt* genes in the *P. rettgeri* strains, we determined the entire nucleotide sequences of the *cdt* gene clusters in the *Providencia* strains (18). Strains for sequencing (*n* = 22) were selected based on their toxigenic activities. These included groups of strains with high toxin titer ≥256 (*P. rustigianii*, 100% of the strains and *P. alcalifaciens*, 25% of the strains), medium titer 32–128 (*P. alcalifaciens*,

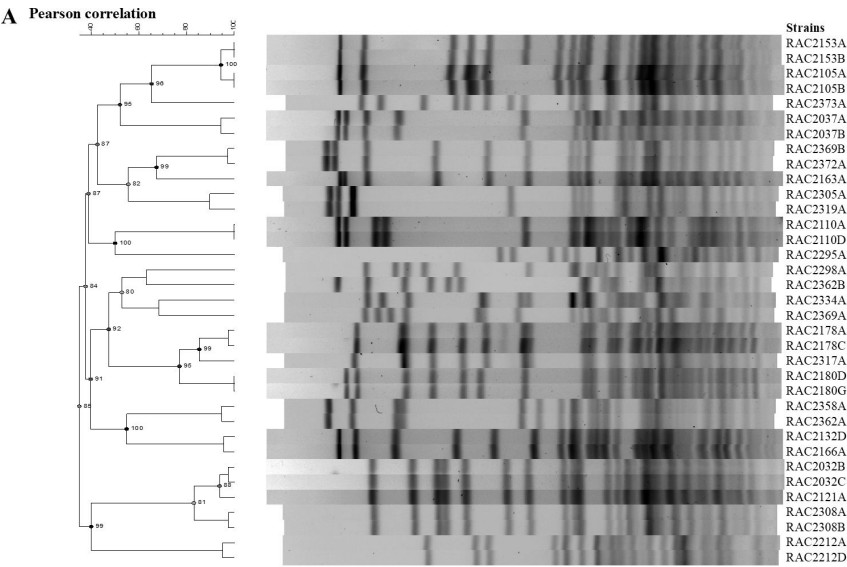

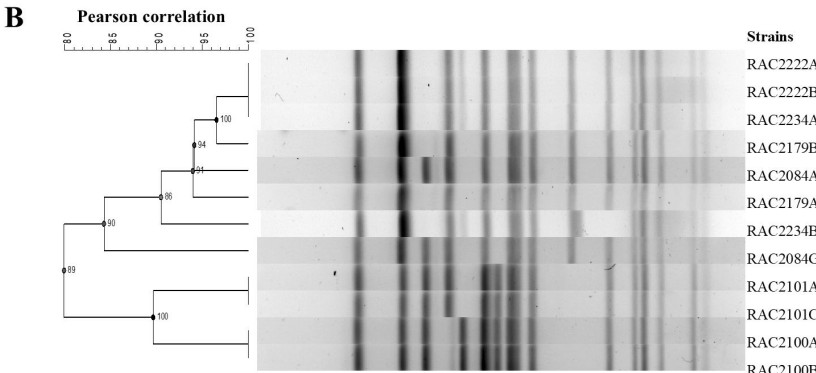

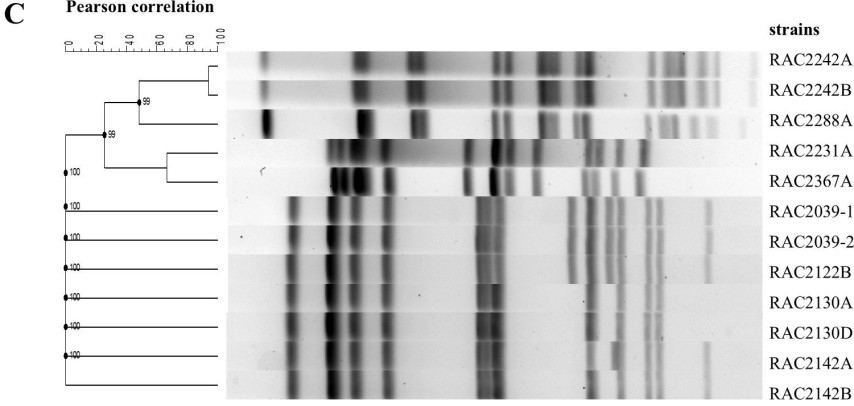

**FIG 1** Phylogenetic analysis of raccoon *cdt* gene-positive *Providencia* strains by PFGE. (A) *Sma*I-digested genomic DNA of 35 *P. alcalifaciens cdt* gene-positive strains isolated from 24 raccoon samples in this study were analyzed by PFGE. One to two isolates were chosen at random from the *cdt* gene-positive samples. The dendrogram was constructed based on DNA fingerprints obtained. The number in each strain name

33% of the strains), low titer 4–16 (*P. alcalifaciens*, 42% of the strains), and strains with no detectable toxin activity (titer ≤1, *P. rettgeri*, 100% of the strains) (Table 4). Complete sequence of the *cdt* gene clusters (*cdtA*, *cdtB*, and *cdtC*) was obtained in all the strains including the *P. rettgeri* strains (Table 4).

Fig 1 (Continued)

represents a specific raccoon identification number. (B) *Not*I-digested genomic DNA of 12 *P. rustigianii* *cdt* gene-positive strains isolated from six raccoon samples in this study were analyzed by PFGE. Two isolates per sample were chosen at random from the *cdt* gene-positive samples. The dendrogram was constructed based on DNA fingerprints obtained. The number in each strain name represents a specific raccoon identification number. (C) *Not*I-digested genomic DNA of 12 *P. rettgeri cdt* gene-positive strains isolated from raccoons in this study were analyzed by PFGE. One to two isolates were chosen at random from the *cdt* gene-positive samples. The dendrogram was constructed based on DNA fingerprints obtained. The number in each strain name represents a specific raccoon identification number.

Analysis of the sequences of *cdt* genes in the *P. rettgeri* strains revealed a conserved CdtB sequence of about 269 amino acids (Table 4) with predicted endonuclease, exonuclease, and phosphatase domains comparable to CdtBs in the *P. alcalifaciens* and *P. rustigianii* (data not shown). However, the *cdtA* and *cdtC* genes were found to be truncated with a premature stop codon in *cdtA*, and deletion of about 141 nucleotides in *cdtC* (Fig. 4). The predicted amino acid sequence of the *P. rettgeri* CdtB has the highest similarity (ca. 95%) to *E. coli* CdtB III/V compared to the known *Providencia* species CdtBs (92%) (Fig. 5). The nucleotide sequences of the *cdt* gene clusters in the *P. alcalifaciens* and *P. rustigianii* strains were devoid of any truncation in the *cdtA*, *cdtB*, and *cdtC* regions (Table 4), and their predicted amino acid sequence alignments were not titer specific as they varied across the high, medium, and low CDT titer strains. However, subtle differences in the amino acid sequences were observed across the *cdt* gene cluster in the *Providencia* strains (data not shown).

## Location of the *cdt* genes in the *Providencia* strains

Since *cdt* genes were not evenly distributed in all the *Providencia* strains, and *Pacdt* and *Pruscdt* genes have been shown to be on a conjugative plasmid (28), we investigated the location of the *cdt* genes in the *cdt* gene-positive *Providencia* strains. S1-PFGE followed by Southern hybridization assay using a $^{32}$P-labeled *PacdtB* gene-probe were conducted. We observed the presence of large plasmids of about 140–210 kb in size that hybridized with the *PacdtB* gene-probe in all the tested *cdt* gene-positive strains (Fig. 6; Fig. S3). Intriguingly, the *PacdtB* gene-probe also hybridized with a smaller plasmid of about 50 kb in size, commonly present in the *P. rustigianii* strains (Fig. 6).

## Presence of plasmid-mediated T3SS in the *cdt* gene-positive *Providencia*

Current advances in the understanding of virulence determinants in *Providencia* species have associated the invasion of human epithelial cells by diarrheagenic *Providencia* strains with the acquisition of a plasmid-mediated T3SS distinct from the chromosomally encoded T3SS that is widely distributed in *Providencia* species (19, 37). Accordingly, the plasmid-T3SS is frequently found co-localized with *cdt* genes on the same plasmid in

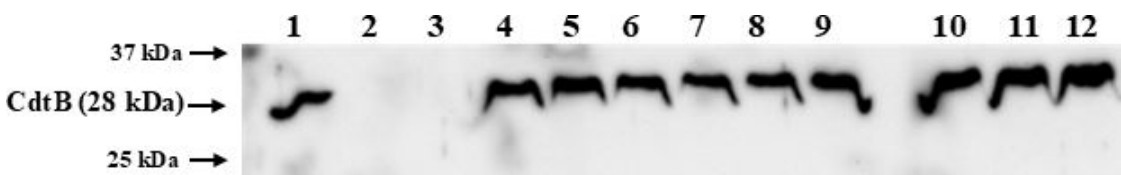

**FIG 2** Detection of CdtB expression by western blotting in the *cdt* gene-positive *Providencia* strains (representative image). 1, *P. alcalifaciens* AH-31 (clinical strain/positive control); 2, *P. alcalifaciens* GTC2020 (negative control); 3, *P. rustigianii* GTC1504 (negative control); and raccoon strains: 4, *P. alcalifaciens* RAC2037A; 5, *P. alcalifaciens* RAC2153A; 6, *P. alcalifaciens* RAC2163A; 7, *P. alcalifaciens* RAC2358A; 8, *P. alcalifaciens* RAC2372A; 9, *P. alcalifaciens* RAC2373A; 10, *P. rustigianii* RAC2084A; 11, *P. rustigianii* RAC2100A; 12, *P. rustigianii* RAC2101A. Cell lysates obtained by boiling at 95°C were separated by SDS-PAGE in a 12% gel, and separated proteins were transferred onto a polyvinylidene difluoride membrane. CdtB was detected by rabbit anti-PaCdtB serum (primary antibody), followed by reacting with anti-rabbit IgG and HRP-linked whole Ab donkey (secondary antibody). Luminescence development was performed using ECL prime western blotting detection reagent.

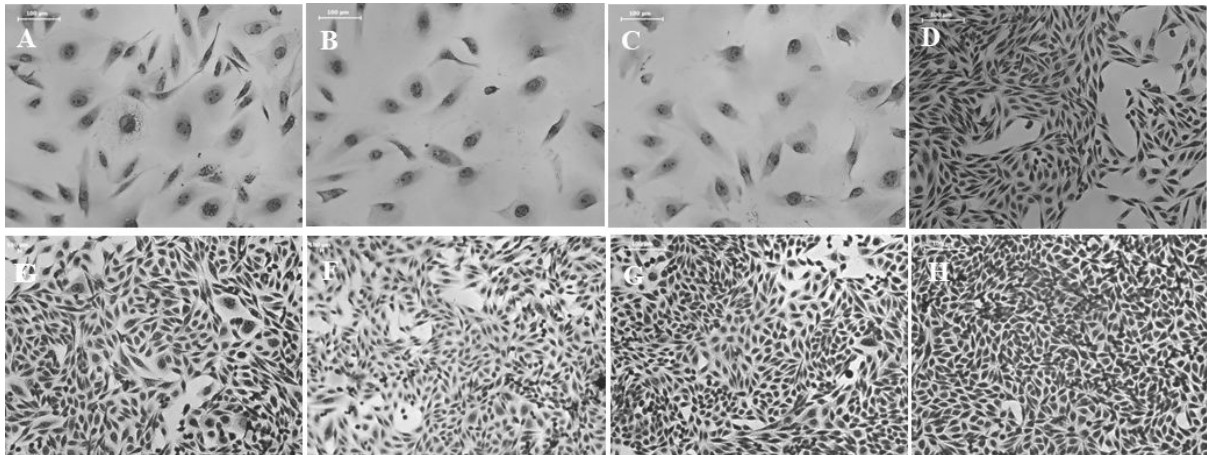

**FIG 3** Effects of *cdt* gene-positive *Providencia* strains cell lysates on CHO cells, magnification ×200 (representative images). CHO cells were treated for 72 h with the filter-sterilized cell lysates from (A) *P. alcalifaciens* AH-31 (clinical strain/positive control); (B) *P. alcalifaciens* RAC2153A (raccoon strain); (C) *P. rustigianii* RAC2084A (raccoon strain); (D) *P. rettgeri* RAC2242A (raccoon strain); (H) *P. alcalifaciens* GTC2020 (*cdt* genes negative/negative control strain). Cell distension activities of the CDTs from the strains (E) *P. alcalifaciens* AH-31, (F) *P. alcalifaciens* RAC2153A, and (G) *P. rustigianii* RAC2084A, were neutralized by the addition of antiserum obtained from rPaCdtB immunized rabbit.

the *cdt* gene-positive *Providencia* strains from diarrheal patients (19). To ascertain the potential of the raccoon-derived *cdt* gene-positive *Providencia* strains to invade human epithelial cells, we screened for the presence of one of the genes (*invF*, a transcriptional regulator of the AraC/XylS family) encoded in the plasmid-borne T3SS genomic island, and distinct from its homolog in the chromosomal-T3SS in the strains. Amplification of the *cspaL* gene (ATP synthetase) was used to evaluate the distribution of chromosomal-mediated T3SS in the *Providencia* strains. The *invF* gene was found to be present in all the *cdt* gene-positive *Providencia* strains (100%, 40/40) and a few *cdt* gene-negative strains (14%, 31/226) indicating the probable presence of the acquired T3SS in the raccoon-derived *Providencia* strains (Table S4). The *cspaL* gene was present in all the *cdt* gene-positive/negative *Providencia* strains (data not shown).

## DISCUSSION

Outbreaks of foodborne illnesses caused by *Providencia* spp. have been reported (5–7), however, the source or reservoir of such virulent *Providencia* is not known. Recent surveillance of wild raccoons in Japan and elsewhere has implicated raccoons as the probable reservoir of some emerging bacterial foodborne pathogens including *E. albertii* (33, 38), *Listeria* spp. (39), and *Salmonella enterica* (34). Consequently, we examined the possible occurrence and distribution of *Providencia* spp. in the wild raccoons in this study. Here, we show that *Providencia* spp. are commonly distributed in wild raccoons and could be the probable source of the virulent *Providencia* strains encoding the *cdt* genes and an additional T3SS on a plasmid, at least in Osaka, Japan. Raccoons may pose a risk to humans either indirectly via transmitting these pathogens to food production

**TABLE 3** Comparison of CDT activities induced by sonic filter-sterilized cell lysates from raccoon and clinical *Providencia* strains post 72 h incubation

| Species (strain) | Origin (n) | Toxin titer CD$_{50}$[a] | | |
|---|---|---|---|---|
| | | CHO cells | Caco-2 cells | HeLa cells |
| *P. alcalifaciens* (AH-31) | Clinical (1) | ×32 | ×32 | ×1 |
| *P. rustigianii* (JH-1) | Clinical (1) | ×32 | ×16 | ×1 |
| *P. alcalifaciens* | Raccoon (40) | ×32 – ×256 | ×32 – ×256 | ×4 – ×128 |
| *P. rustigianii* | Raccoon (12) | ×512 – ×1024 | ×256 – ×1024 | ×128 – ×256 |
| *P. rettgeri* | Raccoon (16) | ×1 | Not tested | Not tested |

[a]Titer CD$_{50}$: the highest dilution of the toxin that distended 50% of the cells.

TABLE 4  *cdt* gene clusters in selected *Providencia* strains

| Species | Toxin titer[a] | Strains | Length (aa) of[b] | | | Accession number |
|---|---|---|---|---|---|---|
| | | | CdtA | CdtB | CdtC | |
| *P. alcalifaciens* | ≥256 (high) | RAC2037A | 280 | 269 | 182 | LC852853 |
| (isolated from | (25%) | RAC2153A | "[c] | " | " | LC852854 |
| 24 samples) | | RAC2305A | " | " | " | LC852855 |
| | 32–128 | RAC2105A | 280 | 269 | 182 | LC852856 |
| | (medium) | RAC2110A | " | " | " | LC852857 |
| | (33%) | RAC2295A | " | " | " | LC852858 |
| | | RAC2372A | " | " | " | LC852859 |
| | 4–16 (low) | RAC2032B | 280 | 269 | 182 | LC852860 |
| | (42%) | RAC2132D | 276 | " | " | LC852861 |
| | | RAC2212A | 304 | " | " | LC852862 |
| | | RAC2308A | 276 | " | " | LC852863 |
| | | RAC2317A | " | " | " | LC852864 |
| | | RAC2358A | " | " | " | LC852865 |
| | | RAC2362A | " | " | " | LC852866 |
| *P. rustigianii* | ≥512 (high) | RAC2084A | 249 | 269 | 182 | LC852867 |
| (isolated from | (100%) | RAC2100A | " | " | " | LC852868 |
| 6 samples) | | | | | | |
| *P. rettgeri* | <1 (no activity) | RAC2039-1 | Truncated | 269 | Truncated | LC852869 |
| (isolated from | (100%) | RAC2130A | " | " | " | LC852870 |
| 8 samples) | | RAC2231A | " | " | " | LC852871 |
| | | RAC2242A | " | " | " | LC852872 |
| | | RAC2288A | " | " | " | LC852873 |
| | | RAC2367A | " | " | " | LC852874 |

[a]Titer: the highest dilution of the toxin that distended 50% of the cells.
[b]aa, Amino acids.
[c] ", Repetition of preceding information in the column.

farms, water reservoirs, or directly when kept as pets or consumed as game meat. In view of this, there is an existing government policy in Japan to curtail the population of raccoons as they are considered nuisance and invasive.

Previous reports (10, 11, 40) have classified *Providencia* strains isolated from disease outbreaks in animals including cats, dogs, and pigs as virulent and potentially diarrheagenic, however, the distributions and transmission dynamics of such strains in those

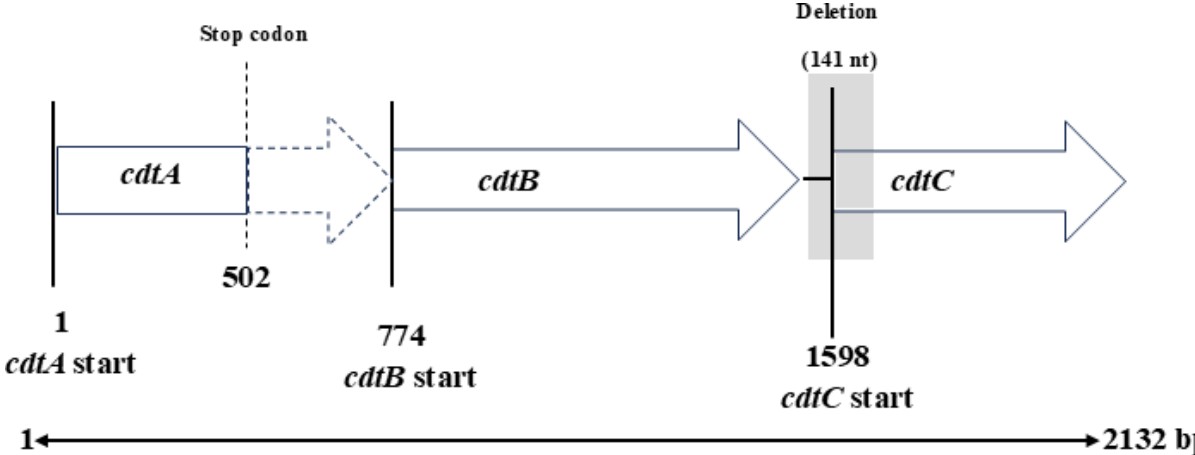

FIG 4  Schematic representation of the *cdt* gene cluster in the *P. rettgeri* strains. Open arrows indicate *cdt* genes ORFs, shaded region indicates the deleted sequence in *cdtC*, broken lines indicate expected incomplete product due to insertion of the nonsense codon. Truncations were observed in *cdtA* and *cdtC*, whereas *cdtB* ORF is devoid of any truncation.

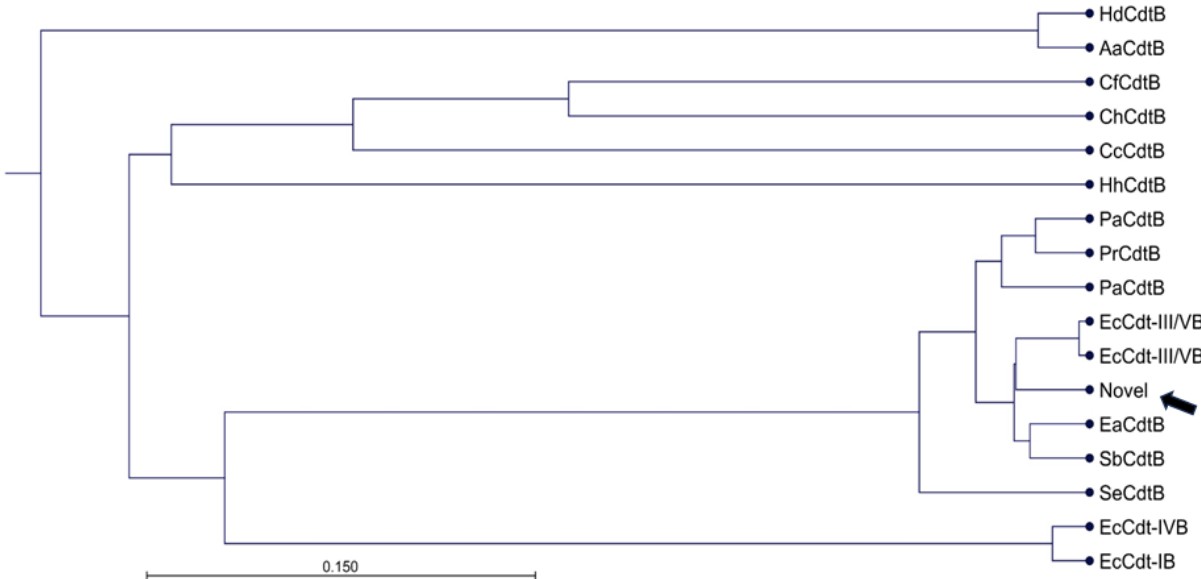

**FIG 5** Evolutionary relationships of the novel *cdt* genes in *P. rettgeri* strains and other *cdt* gene-positive bacteria. The phylogenic tree was constructed with CdtB amino acid sequences of 17 diverse bacterial strains including *P. alcalifaciens* CdtB (PaCdtB: WP_080675667.1, AEM17342.1); *P. rustigianii* CdtB (PrCdtB: BBG92288.1); *E. coli* CdtB (EcCdt: IB, WP_112861021.1; III/V, WP_137548965.1, WP_000759935; IV, EFH5709745.1); *E. albertii* CdtB (EaCdtB: WP_085456383.1); *Salmonella enterica* CdtB (SeCdtB: EBD0958124.1); *S. boydii* CdtB (SbCdtB: AAU88264.1); *Helicobacter hepaticus* CdtB (HhCdtB: AAF19158.1); *Haemophilus ducreyi* CdtB (HdCdtB: WP_010944837.1); *Campylobacter hyointestinalis* CdtB (ChCdtB: WP_147498201.1); *C. fetus* CdtB (CfCdtB: WP_111738196.1); *C. coli* CdtB (CCdtB: EFB0710752.1); *Aggregatibacter actinomycetemcomitans* CdtB (AaCdtB: WP_09497876.1) using UPGMA, with WAG protein substitution model in CLC genomic workbench. The arrow indicates the novel CdtB in *P. rettgeri*.

animals including the asymptomatic ones were not ascertained. Additionally, there are no reports in the literature on the distribution of *Providencia* species including the virulent strains in any wild animal hosts. Our surveillance of the healthy wild raccoons revealed that *Providencia* species (*P. alcalifaciens, P. rettgeri, P. stuartii, P. rustigianii, P. heimbachae, P. vermicola, P. huaxiensis*) are widely distributed in the animals (60%), with the *cdt* gene-positive *Providencia* strains (20%) found in three species including *P. alcalifaciens* (63%), *P. rettgeri* (21%), and *P. rustigianii* (16%). To the best of our knowledge, this is the first report regarding the presence of *Providencia* spp. in raccoons and *cdt* genes in *P. rettgeri*. In addition, the *cdt* gene cluster identified in the *P. rettgeri* strains (Fig. 4) was considered novel based on differences in PCR-RFLP patterns compared to the known *Providencia* species-specific CDTs, e.g., in *P. alcalifaciens* and *P. rustigianii* (3, 18, 22, 28). We observed the co-existence of both *cdt* gene-positive and negative *Providencia* strains in some of the raccoons (data not shown). Though our sampling locations were confined to a limited area in Osaka Prefecture, Japan, we identified multiclonal *cdt* gene-positive *Providencia* strains among the species such as *P. alcalifaciens*, *P. rettgeri,* and *P. rustigianii* circulating in the raccoons, signifying that diverse *cdt* gene-positive strains exist in these animals in Japan and probably elsewhere. Likewise, divergent pulsotypes of the *cdt* gene-positive *Providencia* strains were also found co-existing within the intestinal tracts of some of the raccoons, probably an indication of multi-infection cycles within the raccoon domain. In addition, we also observed that although the animals were usually captured at different locations and at different times in Osaka, Japan, some of the *cdt* gene-positive strains exhibited the same DNA fingerprints (Fig. 1A through C) which could indicate a common source of infection.

The *cdt* gene-positive *Providencia* strains were considered potentially pathogenic to humans as the complete set of toxin genes (*cdtA*, *cdtB*, and *cdtC*) were found in all the strains. However, the toxin genes in the *P. rettgeri* had truncations in the receptor binding subunits (Fig. 4), consisting of a nonsense codon in *cdtA* and deletions in *cdtC*. The same type of mutations at the exact base position and region were observed

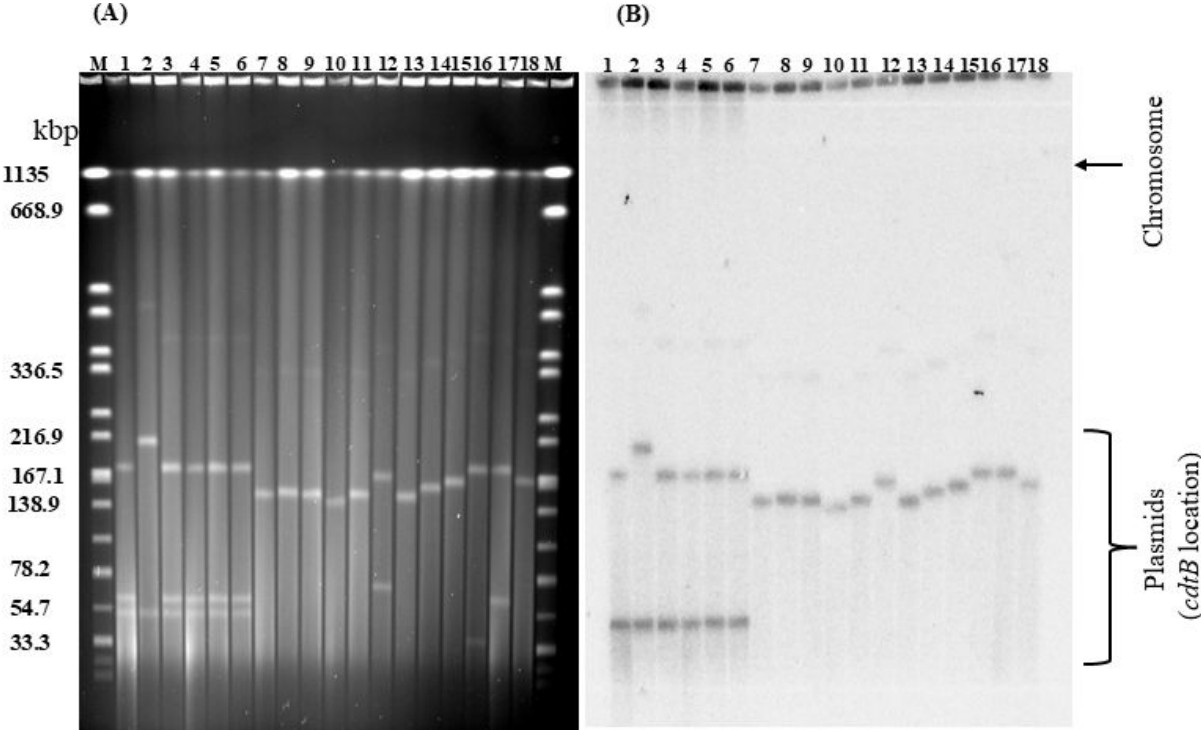

**FIG 6** Location of *cdt* genes in the *Providencia* strains by S1-nuclease PFGE and Southern hybridization (representative image). Genomic DNA of the *Providencia* strains was analyzed by S1 nuclease-PFGE (A) and Southern hybridization assays using $^{32}$P-labeled Pa*cdtB* gene probe (B). Lanes1-6, *P. rustigianii* strains; 7-16, *P. alcalifaciens* strains; 17, *P. alcalifaciens* AH-31 (control 1); 18, *P. rustigianii* JH-1 (control 2); M, molecular size marker (*Salmonella* Braenderup strain H9812 *Xba*I-digested genomic DNA).

in all the *P. rettgeri* strains encoding the *cdt* gene cluster regardless of the clonality. Three nucleotide substitutions both in the forward and reverse primer binding sites in the *P. rettgeri cdt* gene cluster resulted in the inability to be amplified by the duplex PCR specifically designed for this survey. The raccoon-derived *P. rettgeri* isolated in this study could have acquired the toxin genes from a closely related genus *E. coli* (Fig. 5) based on similarities in the nucleotide sequences of these two *cdt* genes. Perhaps, there could be subtle ongoing genetic modifications involving point mutations, deletions, and inversions to assemble a functional toxin cluster within *P. rettgeri* in the raccoons. To further support this assertion, all the *cdt* gene-positive *P. rustigianii* strains have dual *cdtB* genes on different plasmids of varying sizes (Fig. 6). Since this is the first report of two different *cdt* genes present on different plasmids in *cdt*-gene positive bacteria, it would be very interesting to elucidate the nature and importance of the dual toxin clusters in the *P. rustigianii* strains. Importantly, the presence of *cdt* genes in *Providencia* could be a vital virulence biomarker as all the *cdt* gene-positive strains have multiple T3SSs which could enhance their potential to invade and colonize human intestinal epithelial cells. It has been reported that acquired plasmid-mediated T3SS contributes to the invasion of human epithelial cells and intestinal fluid accumulation in the rabbit ileal loop model (19, 37), whereas the widely distributed chromosomal-T3SS in *Providencia* spp. is required for rapid killing of *Drosophila melanogaster* (37). Moreover, the putative virulent determinants (*cdt* gene clusters including the acquired-T3SS) identified among the *Providencia* strains in the current study have the propensity of being transferred to related species due to their locations on plasmids.

The distribution of CDT, a known bacterial virulent determinant, is gradually being unveiled in *Providencia* species. The *cdt* gene-positive *P. alcalifaciens* and *P. rustigianii* clinical strains are known to produce functional CDTs that have been detected immunologically (18, 28). Accordingly, the raccoon-derived *cdt* gene-positive *Providencia* strains

isolated in the present study expressed CdtB indicating the presence of functional toxins in the strains and alluding to their pathogenic fitness. Although we used a primary antibody (CdtB probe) prepared from CdtB of a clinical *P. alcalifaciens* strain AH-31 (18), highly intense bands were observed in the *P. rustigianii* compared to the other tested *Providencia* strains including the clinical and positive control strains. The high intense CdtB reactive bands seen in *P. rustigianii* could be attributed to hyperexpression of the toxin in the strains due to the presence of dual *cdtB* gene on different plasmids. We excluded the possibility of none or differential reactivity of the probe to the targets (CdtBs) as the predicted amino acid sequence of CdtBs in the *Providencia cdt* gene-positive strains were homologous (>85% amino acid sequence similarities). The negligible CdtB band seen in the *P. rettgeri* could be attributed to low or no expression level of the toxin in the strains (data not shown).

Expressed CDTs in the cell lysates of the clinical-derived *cdt* gene-positive *P. alcalifaciens* AH-31 and *P. rustigianii* JH-1 are known to be biologically active in eukaryotic cell lines, though with limited tropism as their activities were restricted to CHO and Caco-2 cells, and not active in HeLa cells (18, 28). Notably, CDTs from the raccoon-derived *P. alcalifaciens* and *P. rustigianii* strains caused extensive cell distension and death of the eukaryotic cell lines compared to the clinical and positive control strains (*P. alcalifaciens* AH-31 and *P. rustigianii* JH-1). Also, CDTs from the raccoon-derived *Providencia* strains seem to have vast cell tropism as they were active on all the tested cell lines including HeLa cells (Fig. 3; Table 3). We do not deny the possibility of low CDT titer in the clinical strains due to long-term storage at −80°C, batch variations of the eukaryotic cell lines compared to the previous studies (18, 28), and the experimental conditions. Nevertheless, some of the raccoon strains showed very high titer against all the tested cell lines. As expected, no distension in any of the tested cell lines was observed with the *cdt* gene-positive *P. rettgeri* strains probably due to the truncations in the CDT receptor binding subunits (CdtA and CdtC). CDT is a tripartite-holotoxin and the expression of the three subunits CdtA, CdtB, and CdtC is indispensable for CDT toxicity (24). CdtA and CdtC are required for binding and internalization of the active subunit (CdtB). It remains to be elucidated whether *P. rettgeri* had any alternative mechanism of utilizing the CdtB in the absence of CdtA and CdtC. For example, *Salmonella* Typhi CDT lacks the receptor binding CdtA and CdtC (41) but utilizes the CdtB only within its host. If the *P. rettgeri* strains are capable of internalizing into their host cells, CdtB devoid of CdtA and CdtC may be enough to cause cell death. Conversely, we do not dismiss the possibility that the newly identified *cdt* gene cluster in the *P. rettgeri* could be a pseudogene cluster with unidentified origin, and probably dispensable for the survival of *P. rettgeri* in raccoons. Likewise, this strain of *P. rettgeri* could have evolved to persist in the raccoons, thereby reducing the evolutionary cost of being pathogenic to its host by converting to an avirulent strain. Nevertheless, further epidemiological studies are warranted to ascertain the distribution of *Pretcdt* genes in this species. Subtle differences in the amino acid sequences of the CDTs could be responsible for the variations in toxin titer levels observed in the *cdt* gene-positive *P. alcalifaciens* strains, as substitutions in amino acid sequences of the receptor binding subunits could lead to reduced binding and toxicity of the holotoxin (42).

In conclusion, this is the first report showing that raccoons could be the reservoir of CDT-producing *Providencia* strains, at least in Japan. In addition, our study has highlighted the zoonotic potential of the *Providencia* species isolated from raccoons including that the *cdt* gene-positive strains were presumed to be more virulent compared to the human clinical strains in terms of CDT production levels. It may be important to survey raccoons from other parts of Japan and elsewhere to explore possible associations with the *cdt* gene-positive *Providencia* strains. For efficient control measures, it may be desirable to widen the scope of this survey to accommodate other sample sources including vegetables and fruits from the farm or environmental water which could be a vital means of contact between the raccoons and humans.

## MATERIALS AND METHODS

### Sample collection

Rectal specimens were collected using cotton swabs (SEEDSWAB γ1, Eiken Chemical Co., Japan) from 384 apparent healthy wild raccoons (*Procyon lotor*) captured in Osaka prefecture, Japan, from November 2020 to April 2023 as part of the Osaka prefectural government extermination and nuisance control programs. The samples were transported to the laboratory at ambient temperature and processed within 6 h of collection.

### Detection and isolation of *Providencia* and *cdt* gene-positive isolates.

One milliliter suspensions of the rectal swabs were made in sterilized Dulbecco's phosphate-buffered saline (PBS) and thereafter, 300 µL of the suspension was inoculated into 3 mL of tryptic soy broth (Becton Dickinson and Co., Franklin Lakes, NJ.) and enriched at 37°C for 14 to 16 h with shaking (180 rpm). Subsequently, 100 µL of the culture was centrifuged at 10,000 $\times$ $g$, 4°C for 3 min. The resulting pellets were resuspended in 85 µL of 50 mM NaOH. The mixture was boiled at 95°C for 10 min, neutralized by the addition of 15 µL of 1 M Tris-HCl buffer (pH 7.0), and centrifuged at 10,000 $\times$ $g$, 4°C for 10 min. The supernatant was subjected to a duplex PCR using two pairs of primers targeting *Providencia* 16S rRNA and *Pcdt* genes, respectively (Table S3). Polymyxin-mannitol-xylitol medium for *Providencia* PMXMP (9) was used in isolating *Providencia* colonies from the PCR-positive samples. Suspected colonies were further confirmed with the duplex PCR specific for *Providencia* genus 16S rRNA and *Pcdt* genes developed in this study.

### Species identification of the *Providencia* isolates

Species of the *Providencia* isolates were determined by conventional biochemical tests (Table S1), and sequencing of 16S rRNA and *rpoB* genes (Table S2) using fresh single colony on LB-agar (Nacalai tesque, INC.), respectively. In the biochemical tests, sugar utility was examined using a basal medium (1% bacto peptone, 0.5% sodium chloride, and 0.0025% bromothymol blue) including one of the following carbohydrates (1% wt/vol): L-arabinose, myo-inositol, rhamnose, sorbitol, trehalose, raffinose, adonitol, or citrate. Alternatively, urea hydrolysis was examined with urea agar medium composed of urea agar base (Thermo Fisher Scientific) and urea (final conc. 2% wt/vol) for the species identification of *Providencia*. The results were interpreted based on the typical properties of each *Providencia* spp. (Table S1) (2, 43–45). The entire 16S rRNA gene sequences and *rpoB* partial sequences of the *Providencia* isolates were amplified as described previously (46, 47) and sequenced as described below. Species identifications were performed with BLAST search (http://www.ncbi.nlm.nih.gov/BLAST) and compared with the biochemical patterns.

### Detection and subtyping of *cdt* genes

A PCR-RFLP assay (22) was used to further examine the presence of other possible *cdt* variant genes which could not be detected with the duplex PCR assay in the isolated *Providencia* strains.

### Amplification of T3SS-associated genes

Specific oligonucleotide primer pairs for plasmid-mediated T3SS-associated *invF* gene and chromosomal-mediated T3SS-associated *cspaL* genes designed based on the dual T3SS sequences of *P. rustigianii* strain JH-1 (19) were used to PCR amplify their respective gene targets.

## Pulsed-field gel electrophoresis

PFGE as described previously (33) was used to ascertain the clonality of the *cdt* gene-positive *Providencia* strains. Briefly, fresh bacterial cells were embedded in an agarose plug and lysed *in situ* to liberate total genomic DNA. The genomic DNA embedded plug was subjected to restriction enzyme digestion with 50 U of *Not*I or *Sma*I (Takara Bio Inc., Shiga, Japan) in 200 µL solution, and electrophoresed in 1% pulsed-field certified agarose (Bio-Rad Laboratories Inc., Hercules, CA) with a CHEF Mapper PFGE (Bio-Rad Laboratories Inc.) using 0.5× TBE buffer (45 mM Tris, 45 mM boric acid, 1 mM EDTA [pH 8.0]). Appropriate run conditions were optimized using the auto algorithm mode of the CHEF Mapper PFGE system for the sizes ranging between 30 and 700 kb for *Not*I and 50 and 300 kb for *Sma*I. *Xba*I-digested genomic DNA of *Salmonella* Braenderup strain H98121 was used as a molecular size marker. The DNA fingerprints of the *cdt* gene-positive strains were interpreted based on Tenover's criteria (48) and analyzed using Bionumerics version 8 software, Applied Maths NV to determine their phylogenetic relationships.

## Western blotting

CDT expression in the *Providencia* strains was examined by Western blotting as described previously (18, 28). Briefly, bacterial cells in LB (Lennox) broth (Nacalai Tesque, INC, Kyoto, Japan) were harvested by centrifugation and adjusted to $OD_{600}$ 5 in PBS. The bacterial suspension was mixed with an equal volume of two times concentrated SDS sample buffer. Then, the mixture was boiled for 10 min, snap-cooled on ice, and separated on 12% SDS-polyacrylamide gel electrophoresis. Separated proteins were blotted onto polyvinylidene difluoride membrane (Bio-Rad Laboratories) with a Trans-Blot SD semidry electrophoretic transfer (Bio-Rad Laboratories). The membrane was blocked with 3% skimmed milk in Tris-buffered saline (TBS: 20 mM Tris-HCl [pH 7.5], 150 mM NaCl) at 4°C for 18 h and probed with 5,000 times diluted anti-rPaCdtB rabbit serum (primary antibody) obtained by immunizing rabbit with recombinant CdtB from a clinical strain of *P. alcalifaciens* (18), at room temperature with shaking for 2 h. After washing the membrane with TBS, the membrane was further incubated with 20,000 times diluted anti-rabbit IgG HRP-linked whole Ab Donkey (Cytiva, Marlborough, MA) as the secondary antibody at room temperature for 2 h with gentle shaking. Luminescence development was performed with Amersham ECL Prime Western blotting detection reagents (Cytiva), and the image was captured using the ChemiDoc Touch Imaging System (Bio-Rad Laboratories).

## Cell culture

HeLa and Caco-2 cells were propagated in minimum essential medium (MEM; Shimadzu Diagnostics Corporation, Tokyo, Japan), whereas CHO cells were cultured in α-MEM (Thermo Fisher Scientific). Media used for HeLa, and other (Caco-2 and CHO) cells were supplemented with 5% and 10% fetal bovine serum (Thermo Fisher Scientific), respectively. All culture media were supplemented with a 1% antibiotic and antimycotic cocktail (Nacalai Tesque, INC, Kyoto, Japan) and 1% GlutaMax (Thermo Fisher Scientific). Also, a 1% nonessential amino acid solution (Thermo Fisher Scientific) was added to the Caco-2 medium. The cells were cultured at 37°C under 5% $CO_2$ in air.

## Cytotoxicity and neutralization assays

The cytotoxicity assay described by Shima et al. (18) was adopted with some modifications. In brief, the *Providencia* strains were cultured in LB broth (Nacalai Tesque, INC) at 37°C, with shaking (180 rpm) for 8 and 16 h, respectively. Bacteria cells propagated for 16 h were collected by centrifugation at 3,800 × *g* at 4°C for 2 min, resuspended in PBS (pH 7.2), and adjusted to $OD_{600}$ 5 whereas filter-sterilized culture filtrates (0.22 µm pore size, Merck Millipore Ltd., Burlington, MA) were obtained from 8 h culture (to check for secretion of CDT in the culture milieus). Cell lysates were prepared by

sonicating the cell suspensions for 1 min on ice using a handy sonicator UR-20P (Tomy Seiko Co. Ltd., Tokyo, Japan). The resulting supernatants from the sonicated lysates were collected by centrifugation at $20,600 \times g$ at 4°C for 15 min, and filter sterilized as described above. Both the filter-sterilized sonic cell lysates and culture filtrates were examined for their CDT activities on the eukaryotic cell lines. *P. alcalifaciens* strain AH-31, *P. rustigianii* strain JH-1, and *E. coli* strain GB1371 were used as positive controls, whereas *cdt* gene-negative *P. alcalifaciens* strain GTC2020, *P. rustigianii* strain GTC1504, and *P. rettgeri* strain GTC1263 were used as negative controls. Ten microliters of the cell sonic lysates or culture filtrates were twofold serially diluted with PBS in a 96-well plate, and 100 µL of MEM or α-MEM containing $5 \times 10^3$ of the eukaryotic cells were added per well and incubated at 37°C under 5% $CO_2$ in air. Morphological changes of the cells were observed under a microscope (Leica DMI6000 B, Leica Microsystems GmbH, Mannheim, Germany) after 72 h incubation, and CDT titer ($CD_{50}$) defined as the highest dilutions of the sonic cell lysates or culture filtrate that caused observable cell distensions of ≥50% of the eukaryotic cells. Multiple (three to six) randomly selected fields of vision were analyzed and compared with the control strains. Each experiment was performed at least thrice to confirm the reproducibility of the data, and fresh sonic lysates or culture filtrates were prepared for every independent experiment.

For the neutralization assay, 10 µL of anti-PaCdtB rabbit serum or pre-immune serum (control) was mixed with 10 µL of $CD_{90}$ dose of filter-sterilized sonic cell lysates or culture filtrates, preincubated at 37°C for 1 h in a 96-well plate. Thereafter, 100 µL of CHO cells ($5 \times 10^3$ cells) were added into each well and incubated at 37°C under 5% $CO_2$ in the air with subsequent observation for cell morphology preservations under a microscope (Leica Microsystems, GmbH) after 72 h incubation.

## Sequencing of the entire *Providencia cdt* gene clusters

The nucleotide sequences of the *cdt* gene clusters in the *Providencia* strains were determined as previously described (18). In brief, PCR amplicons were purified by QIAquick PCR products purification kit (Qiagen, GmbH, Hilden, Germany), and the nucleotide sequence of the amplicons was determined by using a BigDye terminator cycle sequencing kit on an ABI Prism 3100 genetic analyzer (Thermo Fisher Scientific). Synthetic oligonucleotide primers were designed based on obtained sequence templates and genome walking was used to obtain the upstream and downstream sequences of the PCR products. Consensus sequences were generated using the DNA Lasergene software package (DNASTAR Inc., Madison, WI). Sequence homology and phylogenetic tree were constructed using QIAGEN CLC Genomics Workbench 24.0 (QIAGEN).

## S1 nuclease-PFGE and Southern hybridization

The size of the plasmid and location of the *cdt* genes in the *Providencia* strains were examined by S1 nuclease-PFGE (S1-PFGE) and subsequent Southern hybridization with a [32]P-labeled *cdtB* gene-probe from *P. alcalifaciens* AH-31 following the protocol as described previously (28). In brief, fresh bacterial cells were embedded in an agarose plug, lysed *in situ* using cell lysis buffer (10 mM Tris-HCl [pH 7.2], 50 mM NaCl, 100 mM EDTA, 0.2% sodium dodecyl sulfate, 0.5% *N*-lauroylsarcosine) at 70°C for 90 min with gentle shaking. The gel plugs were treated with wash solution (20 mM Tris-HCl [pH 8.0], 50 mM EDTA), with subsequent addition of 5 mL of proteinase K solution (0.8 mg/mL proteinase K [P8044-5G, Sigma Aldrich, Inc., St. Louis, MO], 100 mM EDTA, 0.2% sodium dodecyl sulfate, 1% *N*-lauroylsarcosine [Sigma Aldrich, Inc.]), and gentle shaking at 42°C for 18 h. The genomic DNA in the plugs were treated with 4 U of S1 nuclease (Thermo Fisher Scientific) at 37°C for 45 min in 200 µL of 30 mM sodium acetate buffer (pH 4.6) including 50 mM NaCl, 1.0 mM zinc acetate, and 5% (vol/vol) glycerol. The digested DNA was resolved by PFGE as described above with a switch time of 2.2 to 54.2 s for 20 h. For Southern blotting, the resolved DNA was transferred onto a nylon membrane (Perkin Elmer, Waltham, MA, USA) followed by hybridization with the [32]P-labeled *PacdtB*

gene-probe. Random priming method using MultiPrime DNA Labeling System (Cytiva) with [α-$^{32}$P]-dCTP (111 TBq/mmol) (Perkin Elmer) was used for labeling the *cdtB* gene-probe. BAS FLA-3000 system (Cytiva) was used to visualize radioactivity.

## ACKNOWLEDGMENTS

This study was performed in partial fulfillment of the requirements of a Ph.D. thesis for O.J.O. from the Graduate School of Life and Environmental Sciences, Osaka Prefecture University, Osaka, Japan. O.J.O. was a recipient of the Monbukagakusho (MEXT) Scholarship for a Ph.D. program from the Ministry of Science, Culture, and Sports of Japan.

This work was supported in part by JSPS KAKENHI 20K07483 to S.Y.

## AUTHOR AFFILIATIONS

[1]Graduate School of Life and Environmental Sciences, Osaka Prefecture University, Osaka, Japan
[2]Graduate School of Veterinary Science, Osaka Metropolitan University, Osaka, Japan
[3]Asian Health Science Research Institute, Osaka Metropolitan University, Osaka, Japan
[4]Osaka International Research Center for Infectious Diseases, Osaka Metropolitan University, Osaka, Japan
[5]School of Environment and Life Sciences, Independent University, Dhaka, Bangladesh

## AUTHOR ORCIDs

Okechukwu John Obi  http://orcid.org/0000-0002-1806-3670
Atsushi Hinenoya  http://orcid.org/0000-0003-1149-984X
Sharda Prasad Awasthi  http://orcid.org/0000-0003-1606-3543
Shah M. Faruque  http://orcid.org/0000-0002-7006-622X
Shinji Yamasaki  http://orcid.org/0000-0001-9063-2201

## FUNDING

| Funder | Grant(s) | Author(s) |
| --- | --- | --- |
| MEXT | Japan Society for the Promotion of Science (JSPS) | 20K07483 | Shinji Yamasaki |

## AUTHOR CONTRIBUTIONS

Okechukwu John Obi, Conceptualization, Data curation, Formal analysis, Investigation, Validation, Writing – original draft | Atsushi Hinenoya, Conceptualization, Formal analysis, Supervision, Validation, Writing – original draft, Writing – review and editing | Sharda Prasad Awasthi, Data curation, Formal analysis, Investigation, Methodology, Supervision, Writing – review and editing | Noritoshi Hatanaka, Data curation, Project administration, Supervision, Writing – review and editing | Shah M. Faruque, Formal analysis, Validation, Writing – review and editing | Shinji Yamasaki, Conceptualization, Formal analysis, Funding acquisition, Project administration, Supervision, Writing – original draft, Writing – review and editing

## DATA AVAILABILITY

Nucleotide sequences of the cdt gene clusters obtained from Providencia strains in this study were deposited into the DDBJ database (accession numbers LC852853 to LC852874).

## ETHICS APPROVAL

Approval was obtained from the Osaka Prefectural Government according to the Guidelines for Animal Experimentation of Osaka Prefectural Animal Protection and Livestock Division.

## ADDITIONAL FILES

The following material is available online.

### Supplemental Material

**Supplemental figures (Spectrum02616-24-S0001.pdf).** Figures S1 to S3.
**Supplemental tables (Spectrum02616-24-S0002.docx).** Tables S1 to S3.

### Open Peer Review

**PEER REVIEW HISTORY (review-history.pdf).** An accounting of the reviewer comments and feedback.

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
