## [Reviewer comments · Microbiology Spectrum]

Microbiology Spectrum

Wild raccoons (*Procyon lotor*) as potential reservoir of cytolethal distending toxin producing *Providencia* strains in Japan

Okechukwu Obi, Atsushi Hinenoya, Sharda Awasthi, Noritoshi Hatanaka, Shah Faruque, and Shinji Yamasaki

Corresponding Author(s): Shinji Yamasaki, Graduate School of Veterinary Science, Osaka Metropolitan University

Review Timeline:

Submission Date:	October 16, 2024
Editorial Decision:	November 22, 2024
Revision Received:	December 13, 2024
Accepted:	December 30, 2024

Editor: Blaire Steven

Reviewer(s): Disclosure of reviewer identity is with reference to reviewer comments included in decision letter(s). The following individuals involved in review of your submission have agreed to reveal their identity: Thandavarayan Ramamurthy (Reviewer #1)

Transaction Report:

DOI: <https://doi.org/10.1128/spectrum.02616-24>

Re: Spectrum02616-24 (Wild raccoons (*Procyon lotor*) as potential reservoir of cytolethal distending toxin producing *Providencia* strains in Japan)

Dear Prof. Shinji Yamasaki:

Thank you for the privilege of reviewing your work. Below you will find my comments, instructions from the Spectrum editorial office, and the reviewer comments.

The review for your manuscript were returned and generally favorable. I feel the manuscript will be sufficient for publication after some minor edits.

Please ensure before resubmission that you include a data availability statement and make the sequencing data public.

Revision Guidelines

Sincerely,
Blair Steven
Editor
Microbiology Spectrum

Reviewer #1 (Comments for the Author):

In this manuscript the authors describes about the cytolethal distending toxin producing *Providencia* strains isolated from wild raccoons in Japan. This study was aimed to demonstrate the pathogenic potential of different species of *Providencia* using several phenotypic and genetic markers.

Comments

1. Line 182. The term "expressed" is not aligning well when the test bacterial cells were sonicated and the extracts tested for the biological assay.
2. Lines 173-174. Does the expected size of the CdtB refers to any previous work? If so, provide a reference.
3. Lines 207-212. Mention the specific number of isolates used for sequencing the *cdtB* gene.
4. Is there any difference in nucleotide/amino acid sequences of *cdtB* if found in multiple copies?
5. Line 349-352. Low expression of CDTs could be related to long-term storage of clinical isolates. This needs to be discussed.
6. Does the authors encountered more than one species of *Providencia* spp in a single animal, either by culture or 16s rRNA method.?
7. How the biochemical testing was performed? With the cultured isolates? This has not been mentioned in the methodology. Subsequently, these isolated and identified strains have been used in the PFGE.
8. The authors have used strains and isolates, they differ in biological terms.
9. Lines 95-102. Some of the characteristics of CDT has been reputedly mentioned, e.g. the nature of CDT and its action on human cell-lines.
10. Figure 3. What is the magnification range?
11. Line 269. The meaning of "consumed as games" is not clear.

There were concerns that the English language usage in the manuscript might make it difficult to properly evaluate the science. The ASM Journals webpage provides links to various language editing services (<https://journals.asm.org/writing-your-paper#language-editing-services>). You may consider using these services when revising your manuscript. The use of these services will have no direct bearing on the editorial decision. ASM has no affiliation with these companies.

Response to reviewer's comments

Reviewer #1 (Comments for the Author):

In this manuscript the authors describes about the cytolethal distending toxin producing *Providencia* strains isolated from wild raccoons in Japan. This study was aimed to demonstrate the pathogenic potential of different species of *Providencia* using several phenotypic and genetic markers.

We really appreciate the reviewer for reviewing our manuscript and giving valuable comments. We have carefully checked the comments and would like to reply point-by-point to them as below.

1. Line 182. The term "expressed" is not aligning well when the test bacterial cells were sonicated and the extracts tested for the biological assay.

We deleted “**expressed**” as suggested (Line 186 in the revised MS).

2. Lines 173-174. Does the expected size of the CdtB refers to any previous work? If so, provide a reference.

Reference was added as suggested (28) (Lines 178 in the revised MS).

3. Lines 207-212. Mention the specific number of isolates used for sequencing the cdtB gene.

The number of strains was added (**n = 22**) as suggested (Line 211 in the revised MS).

4. Is there any difference in nucleotide/amino acid sequences of cdtB if found in multiple copies?

We are also interested in knowing that. But we do not have the answer as of now. Since PCR primer which we used for amplification of *cdt* genes could amplify only one *cdt* gene cluster. We are going to analyze second *cdt* genes by WGS.

5. Line 349-352. Low expression of CDTs could be related to long-term storage of clinical isolates. This needs to be discussed.

As suggested, we added the following sentences in Discussion.

We do not deny the possibility of low CDT titer in the clinical strains due to long term storage at -80 C, batch variations of the eukaryotic cell lines compared to the previous studies (18, 28) and the experimental conditions. Nevertheless, some of the raccoon strains showed very high titer against all the tested cell lines. (Lines 359-362 in the revised MS)

6. Does the authors encountered more than one species of *Providencia* spp. in a single animal, either by culture or 16s rRNA method?

Yes. This was confirmed by culture method. We have added the following sentence to the revised MS (Lines 148-150 in the revised MS)

“Mixed *Providencia* species were isolated from 36 raccoons; however, multispecies *cdt* gene-positive *Providencia* were not co-isolated from any of the raccoons”

7. How the biochemical testing was performed? With the cultured isolates? This has not been mentioned in the methodology. Subsequently, these isolated and identified strains have been used in the PFGE.

We have included more detailed information about biochemical testing in the revised MS as follows:

‘using fresh single colony on LB-agar (Nacalai tesque, INC.), respectively. In the biochemical tests, sugar utility was examined using a basal medium (1% bacto peptone, 0.5% sodium chloride and 0.0025% bromothymol blue) including one of the following carbohydrates (1% w/v): L-arabinose, myo-inositol, rhamnose, sorbitol, trehalose, raffinose, adonitol or citrate. Alternatively, urea hydrolysis was examined with urea agar medium composed of urea agar base (Thermo Fisher Scientific) and urea (final conc. 2% w/v) for the species identification of *Providencia*. The results were interpreted based on the typical properties of each

Providencia spp. (Table S1) (2, 44-46)” were added after (Table S2) (Lines 422-430 in the revised MS).

For clarity, in the PFGE section of material and methods, we have modified the first sentence as below.

“isolates” was replaced with “*cdt* gene-positive *Providencia* strains”. (Line 443 in the revised MS).

8. The authors have used strains and isolates, they differ in biological terms.

As advised, we have updated the usage of strains and isolates throughout the manuscript.

9. Lines 95-102. Some of the characteristics of CDT has been reputedly mentioned, e.g. the nature of CDT and its action on human cell-lines.

To make this part clearer, as suggested, we revised as follows (red parts are corrected):

“We have recently found that a type 3 secretion system (T3SS) co-localized with *cdt* genes on a conjugative plasmid distinct from the widely distributed chromosomal T3SSs is linked to invasion of human epithelial cells and diarrheagenicity of *P. rustigianii* (19). Virulence mechanism of CDT produced by *Providencia* spp. remain to be characterized, but association of CDT-producing bacteria from several genera with inflammation and diarrhea in animal models and probably humans have been suggested by previous studies (20 - 23). (Lines 91 to 97 in the revised MS).

10. Figure 3. What is the magnification range?

As shown in Figure 3, the images have uniform magnifications of x100 μ m. For clarity, we have added “magnification x100 μ m” at the figure legend.

11. Line 269. The meaning of "consumed as games" is not clear.

“consumed as games” was replaced with “consumed as game meat” in the revised MS (Line 273 in the revised MS).

Additional revision

We have deposited *cdt* gene sequences obtained in this study into public database (DDBJ) and got their accession numbers. Accordingly, the following description below was added in the revised MS (L554-556). The numbers were also added to Table 4.

Nucleotide sequence accession numbers. Nucleotide sequences of the *cdt* gene clusters obtained from *Providencia* strains in this study were deposited into the DDBJ database (accession numbers LC852853 to LC852874).

Re: Spectrum02616-24R1 (Wild raccoons (*Procyon lotor*) as potential reservoir of cytolethal distending toxin producing Providencia strains in Japan)

Dear Prof. Shinji Yamasaki:

I would like to apologize for any confusion in the review process. The manuscript has now received two reviews and has been recommended for publication.

Thank you for your patience and submitting your work to Spectrum.

Your manuscript has been accepted, and I am forwarding it to the ASM production staff for publication. Your paper will first be checked to make sure all elements meet the technical requirements. ASM staff will contact you if anything needs to be revised before copyediting and production can begin. Otherwise, you will be notified when your proofs are ready to be viewed.

Sincerely,
Blair Steven
Editor
Microbiology Spectrum

Reviewer #1 (Comments for the Author):

None